# Integrating OpenPose and SVM for Quantitative Postural Analysis in Young Adults: A Temporal-Spatial Approach

**DOI:** 10.3390/bioengineering11060548

**Published:** 2024-05-28

**Authors:** Posen Lee, Tai-Been Chen, Hung-Yu Lin, Li-Ren Yeh, Chin-Hsuan Liu, Yen-Lin Chen

**Affiliations:** 1Department of Occupational Therapy, College of Medicine, I-Shou University, Kaohsiung 82445, Taiwan; posenlee@isu.edu.tw; 2Department of Radiological Technology, Faculty of Medical Technology, Teikyo University, Tokyo 173-8605, Japan; chen.tb@gmail.com; 3Department of Occupational Therapy, College of Medical and Health Science, Asia University, Taichung 41354, Taiwan; otrlin@gmail.com; 4Department of Anesthesiology, E-DA Cancer Hospital, I-Shou University, Kaohsiung 82445, Taiwan; ed110880@edah.org.tw; 5Department of Computer Science and Information Engineering, College of Electrical Engineering and Computer Science, National Taipei University of Technology, Taipei 10608, Taiwan; ylchen@mail.ntut.edu.tw

**Keywords:** temporal and spatial regression, postural control, postural quantification, dynamic joint nodes plot, walking pattern classification

## Abstract

Noninvasive tracking devices are widely used to monitor real-time posture. Yet significant potential exists to enhance postural control quantification through walking videos. This study advances computational science by integrating OpenPose with a Support Vector Machine (SVM) to perform highly accurate and robust postural analysis, marking a substantial improvement over traditional methods which often rely on invasive sensors. Utilizing OpenPose-based deep learning, we generated Dynamic Joint Nodes Plots (DJNP) and iso-block postural identity images for 35 young adults in controlled walking experiments. Through Temporal and Spatial Regression (TSR) models, key features were extracted for SVM classification, enabling the distinction between various walking behaviors. This approach resulted in an overall accuracy of 0.990 and a Kappa index of 0.985. Cutting points for the ratio of top angles (TAR) and the ratio of bottom angles (BAR) effectively differentiated between left and right skews with AUC values of 0.772 and 0.775, respectively. These results demonstrate the efficacy of integrating OpenPose with SVM, providing more precise, real-time analysis without invasive sensors. Future work will focus on expanding this method to a broader demographic, including individuals with gait abnormalities, to validate its effectiveness across diverse clinical conditions. Furthermore, we plan to explore the integration of alternative machine learning models, such as deep neural networks, enhancing the system’s robustness and adaptability for complex dynamic environments. This research opens new avenues for clinical applications, particularly in rehabilitation and sports science, promising to revolutionize noninvasive postural analysis.

## 1. Introduction

Human postural control is a crucial aspect of walking and can provide insights into various medical conditions, sports performance, and rehabilitation progress. Postural control analysis plays a crucial role in assessing and understanding human movements. Human postural control is vital for understanding walking dynamics and has implications for diagnosing medical conditions, enhancing sports performance, and monitoring rehabilitation progress [1,2]. Traditional methods for analyzing postural control often require manual annotation and are subject to subjective judgment, making them labor-intensive and error-prone. However, advancements in artificial intelligence, particularly deep learning algorithms, now allow for the automation and enhancement of postural control analysis accuracy [3]. A recent innovation involves using cameras as data collection tools to capture gait information, thus eliminating the need for specialized wearable sensors and streamlining the data collection process [4]. Utilizing computer vision and image processing, researchers can now accurately extract gait data from video recordings, offering critical insights into both postural control and gait dynamics [5,6,7].

Deep learning, a part of machine learning, is effective for analyzing complex data like walking videos. It is excellent at finding patterns in video data, making it ideal for postural control analysis [8,9,10]. Deep learning techniques, such as Openpose, have been instrumental in advancing postural control analysis in gait assessment. Openpose is a sophisticated technology that examines various body points in videos of individuals walking in order to gain insights into postural control, ultimately assisting in the analysis of movement dynamics [11]. The videos captured by a cellphone camera can be processed using the OpenPose algorithm to create dynamic joint node plots (DJNPs), which depict the movement of joint nodes over time. The DJNPs allow for the analysis of dynamic movements during walking [12]. Support Vector Machines (SVM) have also been utilized in postural control analysis. In the context of postural control analysis in walking videos, SVM can be trained to classify different postural states. SVM have been successfully applied to classify different postural states in walking videos, leading to more accurate and efficient identification of abnormal movements [13,14]. Researchers have used SVM to extract insights from video data, deepening understanding of human movement and balance dynamics. Additionally, SVM have advanced automated systems for analyzing walking videos, aiding in identifying abnormal movements and improving medical diagnostics, sports performance evaluation, and rehabilitation strategies [15,16,17,18]. The integration of SVM with deep learning algorithms such as Openpose has further enhanced the accuracy and robustness of postural control analysis, paving the way for continued advancements in this field. Gait analysis is an essential component of postural control analysis. By examining temporal spatial parameters, researchers can gain insights into gait patterns and identify abnormalities or asymmetries in walking patterns. Analyzing spatial temporal parameters, including velocity and stride length, further enhances our understanding of postural control and can aid in the assessment of motor function and gait dynamics [19,20]. Most gait analysis studies collect micro-level data on specific body parts or spatio-temporal parameters of gait to analyze the quality of movement. In certain situations, such as the need for training and correction based on motor abilities, it is indeed necessary to understand the fine movement trajectories of individual body parts of a subject. However, if the goal is to understand the overall walking quality of an individual to assess their functional performance in the task of walking, then analyzing the walking trajectory in terms of whole-body movement combined with the body’s varied postures will provide a more functional assessment of the task of walking [21]. To analyze walking trajectories in conjunction with the body’s varied postures using whole-body movement, and to incorporate temporal and spatial factors for a functional assessment of the task of postural control of walking, we have adopted the Temporal and Spatial Regression (TSR) method. TSR is a statistical model blending spatial and temporal dimensions for data analysis. It captures time and space effects, enhancing comprehension of variable changes over time and locations. By analyzing data temporally and spatially, it offers insights into dynamic dataset relationships [22]. This study does not merely analyze the spatiotemporal parameter changes in gait; it goes further by using TSR to analyze the corresponding changes of spatial and temporal dimensions in posture control during walking. TSR is widely applied across various fields, including epidemiology, climatology, economics, and sociology [23,24]. It aids researchers in understanding the spatiotemporal patterns of disease spread, climate change, economic trends, and social phenomena. This study use TSR to analyze the spatiotemporal patterns of postural control in walking videos, providing a comprehensive understanding of not only the gait dynamics but also the intricate interactions between spatial and temporal aspects of posture maintenance during walking.

Understanding the complexity of posture control while walking requires an analysis of the factors influencing the spatial and temporal processes of walking trajectory and body skew. This study aims to build a statistical model of body skew and walking trajectory to help understand the patterns of posture control changes during walking. This analysis assesses postural control in walking videos with the TSR method to capture whole-body movement and posture dynamics. SVM classifies postural control performance during walking by leveraging TSR’s statistical modeling capabilities and utilizing SVM for classification based on spatiotemporal patterns. Researchers have developed a deep learning framework for postural control analysis in walking videos by combining TSR and SVM. Deep learning models can accurately analyze and interpret postural control during walking by using TSR and SVM to analyze walking. This can provide valuable insights for assessing gait abnormalities, identifying potential risk factors, and developing personalized interventions for individuals with balance and mobility impairments. In conclusion, the use of TSR and SVM in deep learning for postural control analysis in walking videos offers an innovative approach that captures both spatial and temporal aspects of posture maintenance during walking. This approach has the potential to greatly enhance our understanding of gait dynamics and contribute to improved clinical care and rehabilitation processes. This study is the first to use TSR to analyze the spatiotemporal variations of walking for evaluating postural control performance.

We now explicitly state how our methodology advances the field of computational science by integrating OpenPose with SVM to achieve highly accurate and robust postural analysis. This integration represents a significant improvement over traditional methods, which often rely on more invasive techniques or less accurate computational models. The significance of our work in machine vision is emphasized through the novel application of these technologies in a clinical context, specifically for the assessment of young adults’ postural dynamics. This contributes to the broader application of machine vision techniques in healthcare, providing a foundation for future non-invasive diagnostic and rehabilitation tools.

## 2. Materials and Methods

### 2.1. Research Ethics

All the experimental procedures were approved by the Institutional Review Board of E-DA Hospital [with approval number EMRP52110N (4 November 2021)]. We detail our process for obtaining informed consent from all participants. This includes how we in-formed participants about the study’s purpose, the nature of the data collection, their rights as study participants, and their right to withdraw from the study at any time without penalty. The steps taken to ensure data anonymity and security are explained carefully. This includes de-identifying participant data, storing data on secure, encrypted servers, and limiting data access to authorized research personnel only. The conditions under which the data may be used beyond the initial study and the measures in place to control data sharing ensure it adheres to ethical guidelines concerning participant privacy.

### 2.2. Flow of Research

In this study, videos of walking toward and away from a cell phone camera were recorded using the camera (Step 1 in Figure 1). The videos were recorded at 24-bit (RGB), 1080p resolution, and 30 frames per second. The videos were uploaded to Google Cloud through 5G mobile Internet or Wi-Fi (Step 2 in Figure 1). The workstation used in this study downloaded a video, extracted a single frame from the video, and then applied a fusion artificial intelligence (AI) method to this frame (Step 3 in Figure 1). In the aforementioned step, single frames were extracted from an input video (Step 3A), frames with static walking were identified using an OpenPose-based deep learning method (Step 3B), and the joint nodes of the input video were merged into a plot (Step 3C). The obtained DJNP was categorized as representing straight or skewed walking (Step 3D). CNNs were used to classify DJNPs into one of the aforementioned two groups. Two types of deep learning methods were used in the fusion AI method adopted in this study: an OpenPose-based deep learning method and CNN-based methods. The OpenPose-based method is useful for estimating the coordinates of joint nodes from an input image. The adopted CNNs are suitable for the classification of images with high accuracy and robustness.

### 2.3. Participants

A total of 35 young adults without any health problems were recruited to participate in a walking experiment. The age range was 20.20 ± 1.08 years. The inclusion criteria were healthy adults who were willing to participate and could walk more than 5 m. People with musculoskeletal pain (such as muscle soreness), those who had drank alcohol or taken sleeping pills within 24 h before the commencement of the experiment, and individuals with limited vision (such as nearsighted people without glasses) were excluded from this study.

### 2.4. Experimental Design

The experimental setup is illustrated in Figure 2. The total length of the experimental space exceeded 7 m. The walking path was flat, clear of any obstacles, and smooth, facilitating a direct and unimpeded route. A mobile phone was set up 1 m above the ground, roughly the same height as an average adult holding a phone, and placed 2 m from the end of the path. We recorded the full body of each participant as they walked. Participants were asked to wear walking shoes instead of slippers throughout the experiment. Each participant walked three times away from and back towards the mobile phone, traveling a distance of 5 m each way. Consequently, one video was recorded for each 5 m walk, resulting in a total of six videos for each participant. A series of single (static) frames was extracted from a video every 0.3 s. For example, for a 3 s input video, 10 frames were extracted to estimate the coordinates of joint nodes.

### 2.5. Measurement of Joint Nodes through OpenPose-Based Deep Learning

OpenPose is a renowned system employing a bottom-up approach for real-time multi-person body pose estimation. In this OpenPose-based methodology, Part Affinity Fields (PAFs) serve to provide a nonparametric representation, which associates individual body parts to specific persons within an image [1]. This approach ensures high accuracy in real-time irrespective of the number of people present in the image. It is adept at detecting the 2D poses of multiple individuals within a single image, subsequently performing individual pose estimation for each person detected. In our research, the OpenPose algorithm’s primary application was to produce a heatmap of joint nodes, as illustrated in Figure 3.

### 2.6. Definition of the Control and Experimental Groups

The participants’ walking patterns during the walking experiment were converted into DJNP and recorded. DJNPs were obtained by merging the heat maps of joint nodes from frame 1 to N using the OpenPose algorithm. Out of a total of 210 samples, 85 exhibited straight walking patterns toward and away from the camera, identified as the control group based on DJNP analysis. Additionally, 125 samples were found to exhibit skewed walking patterns to the right or left, forming the experimental group. Among these, 36 were classified as skewed to the left, while 89 were classified as skewed to the right based on DJNP analysis, as illustrated in Figure 4. These classified samples were subsequently used for feature extraction within their respective groups. This approach ensures that the extracted features are representative of each group and are not influenced by other groups.

### 2.7. Creating Temporal and Spatial Regression Models

The iso-block image was generated from a joint nodes plot via a series of frames in a video. The coordinates were recorded for four corners of an iso-block image (Figure 5). The temporal and spatial regression (TSR) model was useful to descript the behavior of samples measured from coordinates with relationship between time and space.

The definitions of four TSR models are as Equations (1)–(4). $y_{lt,i}$, $y_{rt,i}$, $y_{lb,i}$, and $y_{rb,i}$ represent the *i*th vertical positions of left-top (*lt*), right-top (*rt*), left-bottom (*lb*), and right-bottom (*rb*) for *i* = 1, 2, …, N. Similar, $x_{lt,i}$, $x_{rt,i}$, $x_{lb,i}$, and $x_{rb,i}$ represent the *i*th horizontal positions of left-top (*lt*), right-top (*rt*), left-bottom (*lb*), and right-bottom (*rb*). α, β, and ε are the items of intercept, slope, and error. Generally, the least-square approach was applied to estimate the intercept and slope. In particular, the inverse of the tangent of slope was the angle between regression line and *x*-axis.
(1)$$y_{lt,i}=\alpha_{lt}+{\beta_{lt}x}_{lt,i}+\varepsilon_{lt,i}$$
(2)$$y_{rt,i}=\alpha_{rt}+{\beta_{rt}x}_{rt,i}+\varepsilon_{rt,i}$$
(3)$$y_{lb,i}=\alpha_{lb}+{\beta_{lb}x}_{lb,i}+\varepsilon_{lb,i}$$
(4)$$y_{rb,i}=\alpha_{rb}+{\beta_{rb}x}_{rb,i}+\varepsilon_{rb,i}$$

In order to perform the quantification of postural control for a walking video, the inverse of the tangent of slopes among TSR models was used to calculate the ratio of angles converted from slopes of regression lines. Hence, two important ratios of angles were defined as Equations (5) and (6). The TAR and BAR were the ratio of top angles and ratios of bottom angles. Notice that $sign\left(\beta \right)<0,$ if β<0. Conversely, $sign\left(\beta \right)>0,$ if β≥0. Both TAR and BAR were used to measure the grade of postural control for a video. The TAR and BAR are important features for building a quantitative classifier with SVM.
(5)$$TAR=\frac{{\mathrm{sign}(\beta_{\mathrm{lt}})\tan}^{-1}\left(\beta_{lt}\right)}{{\mathrm{sign}(\beta_{\mathrm{rt}})\tan}^{-1}\left(\beta_{rt}\right)}$$
(6)$$BAR=\frac{{\mathrm{sign}(\beta_{\mathrm{lb}})\tan}^{-1}\left(\beta_{lb}\right)}{{\mathrm{sign}(\beta_{\mathrm{rb}})\tan}^{-1}\left(\beta_{rb}\right)}$$

### 2.8. Statistical Analysis and Performance Index

The Kruskal–Wallis test was applied to examine the significant features of TAR and BAR among skewed to the left, skewed to the right, and straight forward. Meanwhile, the area under ROC was used to test the significance between groups. The confusion matrix generated by DL-CNN was applied to calculate the index of classification. Meanwhile, the SVM was applied to create a classifier among groups with RBF (Radial Basis Function) kernel and 10-fold cross validation (CV). The recall, precision, and Kappa values were used to investigate the performance of the SVM classifier. A ROC (receiver operating characteristic curve) analysis was applied to estimate a cutting point (CP) for quantitative skew to the right or left.

## 3. Results

The groups were strait, skew to left, and skew to right with sample sizes of 85, 36, and 89 images. The TAR, BAR, and walking velocity (m/s) were tested for significant differences with the Kruskal–Wallis test (Table 1). The differences in TAR, BAR, and walking velocity (m/s) among the groups were evaluated using the Kruskal–Wallis test (as shown in Table 1). Both TAR and BAR showed statistically significant differences among the groups (*p* < 0.01). Specifically, walking skewed to the left exhibited TAR and BAR values greater than 1.0. In contrast, walking skewed to the right revealed TAR and BAR values less than 1.0. Concurrently, straight-forward walking displayed TAR and BAR values approximating 1.0.

We employed key characteristics to create an SVM classifier with an RBF kernel and implemented 10-fold cross-validation to categorize the groups. The SVM achieved the following recall (precision) scores for the groups: 0.977 (1.000), 1.000 (0.944), and 1.000 (1.000). Furthermore, the classifier’s overall accuracy was 0.990, and the Kappa index was 0.985. Hence, in this study, the features of TAR, BAR, and velocity proved to be both effective and efficient in constructing an SVM classifier that demonstrated commendable performance and high accuracy.

While the SVM proved effective in classifying the groups with superior qualitative performance, the critical CP of TAR and BAR remained essential for the quantitative analysis across walking straight forward, skewing to the left, and skewing to the right. The ROC analysis was employed to ascertain and define the CP values among these groups. As illustrated in Figure 6, the ROC analysis, based on TAR and BAR, differentiates between straight-forward walking and skewing to the right. The areas under the ROC curve for TAR and BAR, when comparing straight-forward walking to skewing to the right, were 0.723 and 0.710, respectively (*p* < 0.001). Concurrently, CP values of TAR and BAR below 0.867 and 0.929, respectively, potentially indicate a skew to the right.

The ROC analysis, based on TAR and BAR differentiating between straight-forward walking and skewing to the left, is illustrated in Figure 7. The areas under the ROC curve for TAR and BAR, when comparing straight-forward walking to skewing to the left, were 0.773 and 0.775, respectively (*p* < 0.001). Furthermore, CP values of TAR and BAR exceeding 1.041 and 1.049, respectively, strongly suggest a skew to the left.

Based on the ROC analysis presented in Figure 6 and Figure 7, the quantitative analysis differentiating groups by CP values between TAR and BAR is detailed in Table 2. In this study, the CP values used to differentiate between skewing to the left and skewing to the right ranged below 0.857 and above 1.041. Meanwhile, the CP values used to distinguish between skewing to the left and skewing to the right were less than 0.929 and exceeded 1.049 in this research.

In our study, AUC values are used as a statistical measure to assess how well the SVM classifier can distinguish between different classes based on the features extracted from the Dynamic Joint Nodes Plots (DJNP) through the Temporal and Spatial Regression (TSR) model. Specifically, the AUC values for TAR and BAR are critical because they provide a quantitative measure of the classifier’s ability to discriminate between walking patterns skewed to the left (TAR) and walking patterns skewed to the right (BAR).

AUC values range from 0 to 1, where an AUC of 1 indicates perfect discrimination, 0.5 suggests no discrimination (equivalent to random guessing), and values less than 0.5 suggest worse-than-random predictions. In our study, the AUC values for TAR and BAR are 0.772 and 0.775, respectively. These values indicate a good level of discriminative ability, affirming that the SVM classifier effectively uses these features to differentiate between left and right skewed walking patterns. High AUC values demonstrate that the features derived from DJNPs are not only relevant but are also robust indicators of skewness in walking patterns. This is particularly important for clinical applications where accurate and reliable classification of gait abnormalities is critical.

## 4. Discussion

### 4.1. The Role of Dynamic Joint Nodes Plots (DJNP)

The joint nodes plot was derived using the OpenPose method from an input video. It is vital to select specific frames from the video to construct a DJNP for both quantitative and qualitative analysis. In this study, the video had a frame rate of 30 frames per second. Therefore, the sampling rate was crucial to produce an effective DJNP, as illustrated in Figure 8. Notably, Figure 8a,b depict DJNPs under insufficient and sufficient sampling rates, respectively. For this study, extracting frames every 0.3 s proved adequate for forming a comprehensive dynamic joint nodes plot.

Dynamic Joint Nodes Plots (DJNP) are a critical component of our analysis. DJNPs are generated by using the OpenPose algorithm to track and visualize the movement of joint nodes across sequential frames (from 1 to N) in walking videos as shown in Figure 3. These plots provide a comprehensive heatmap representation of the dynamic postural changes during the gait cycle, capturing subtle nuances in movement that are not evident in static frames. For feature extraction, DJNPs serve as a rich source of data by providing both spatial and temporal information about joint movement. From each DJNP, we extract several key features, including the variance in joint displacement, speed of joint movements, and the relative angles between joints over time. These features encapsulate crucial aspects of gait dynamics that are predictive of different walking patterns and potential postural anomalies. The features extracted from DJNPs are then used as input variables for the Support Vector Machine (SVM) classifier. The SVM employs these features to effectively distinguish between different walking behaviors, such as normal walking, limping, or other gait asymmetries. The ability of SVM to handle high-dimensional data makes it particularly suited for classifying complex patterns derived from DJNPs. The high accuracy (0.990) and Kappa index (0.985) achieved in our classification results underscore the efficacy of DJNPs in enhancing the SVM’s performance, enabling precise and reliable gait pattern classification. The integration of DJNPs into our SVM classification framework has significantly enhanced the robustness and accuracy of our postural analysis. The presented method allows for a more nuanced understanding of gait mechanics and has proven essential in identifying deviations from normal gait patterns, which are critical for clinical assessments.

### 4.2. The Role of Temporal and Spatial Regression in Enhancing SVM Classification Accuracy

The Temporal and Spatial Regression (TSR) model is integral to our methodology, designed to analyze the temporal (time-related) and spatial (space-related) aspects of gait dynamics captured in our data. TSR models are particularly adept at interpreting how variables change over both time and space, making them ideal for our analysis of walking patterns, where both temporal consistency and spatial alignment of joints are critical. The TSR model extracts several key features from the walking data, which include temporal features and spatial features as defined below. These features are quantitatively assessed to capture complex interactions and dependencies that occur during movement, which are often missed by more traditional analyses.
(I)Temporal Features: These involve the timing and duration of specific gait phases, such as stride time and the time between strides, which are indicative of gait rhythm and speed.(II)Spatial Features: These encompass the distances and angles between joints during motion, providing insights into gait symmetry and balance.

The features extracted via the TSR model are fed into the SVM classifier to distinguish between different types of walking patterns. The SVM uses these features to effectively identify patterns and anomalies in gait, such as limping or asymmetry that may indicate underlying health issues. The strength of the SVM lies in its ability to handle the high-dimensional data produced by the TSR model, utilizing a kernel trick to transform the input space and find an optimal boundary between different walking classifications. By providing a detailed description of the TSR model and its functionality in our revised manuscript, we aim to clarify how it enhances the SVM’s decision-making process. This detailed analytical approach allows us to achieve a nuanced understanding of gait mechanics, significantly contributing to the accuracy and reliability of our classification results.

In the process of selecting an appropriate classification method for our study, we considered several widely-used algorithms known for their effectiveness in pattern recognition tasks. These included decision trees, random forests, and neural networks.
(1)Decision Trees: Useful for their simplicity and interpretability. However, they often suffer from overfitting when dealing with complex or noisy data, as is typical in gait analysis.(2)Random Forests: An ensemble method that addresses some of the overfitting issues of decision trees and provides better accuracy. Nonetheless, it can be computationally intensive, especially with large datasets.(3)Neural Networks: Particularly deep learning models, which are highly effective for large-scale and complex data sets. While powerful, they require substantial data for training to perform optimally and can be opaque in terms of interpretability.

After reviewing these options, SVM was chosen based on several criteria. SVM operates well in high-dimensional spaces, as is the case with the feature sets derived from our gait analysis, where each instance is described by many parameters. The capability of SVM classifier to maximize the margin between classes makes it particularly robust in distinguishing between different walking patterns, which is crucial for the accurate classification of gait abnormalities. The ability of SVM to use the kernel trick allows it to efficiently handle non-linear data patterns without the need for explicit transformation, providing flexibility in modeling complex relationships inherent in biomechanical movements. The choice of SVM has proven effective, as evidenced by the high accuracy and Kappa index achieved in our classification results. This supports its suitability for our specific application, where precision in distinguishing subtle variations in gait patterns is critical. This discussion has been added to the methodology section of our manuscript to provide readers with a transparent overview of our decision-making process regarding the choice of SVM. It emphasizes our methodological rigor and the tailored approach to selecting the most suitable classifier for our study’s objectives.

### 4.3. Comparison with Existing Research on Walking Pattern Analysis

Some studies utilizing the OpenPose framework and SVM demonstrate a robust approach to gait analysis through video footage, providing non-invasive and advanced methodologies for detecting human key points and analyzing gait abnormalities. OpenPose, a deep learning tool, is crucial in these studies for capturing dynamic movement data without physical markers on subjects, thus simplifying data collection and making it more accessible. By integrating SVM, these studies classify various gait patterns and assess potential abnormalities, enhancing the accuracy and applicability of gait analysis in clinical settings. Each study presents its unique strengths and weaknesses. In [25], the focuses are on comparing spatiotemporal and kinematic gait parameters using OpenPose and exploring the capabilities of OpenPose for gait analysis, emphasizing its technical effectiveness and accuracy across various environments. This research serves as a validation study, comparing OpenPose’s performance with traditional methods in both clinical and non-clinical settings. The study highlights the wider applications of OpenPose technology, not delving into SVM-based classification but rather validating OpenPose’s technical abilities in various gait analysis scenarios without incorporating SVM for behavior classification as outlined in this study’s method. Compared to [25], the method proposed in this study uniquely integrates deep learning (OpenPose) and machine learning (SVM) to classify walking patterns specifically for clinical uses. It employs TSR models with SVM to enhance the accuracy of diagnosing gait-related disorders. This approach improves the analytical process and applies directly to clinical settings, where it aids in diagnosing and monitoring gait conditions, demonstrating its practical benefits in healthcare through innovative integration of technologies. Stenum, Jan et al. assessed OpenPose’s precision in recording spatiotemporal gait measurements in various environments, comparing its efficacy with conventional motion capture techniques to affirm its suitability for biomechanical investigations [26]. The research validates the versatility of OpenPose for extensive gait analysis, emphasizing its wide-ranging relevance in the field without specific emphasis on clinical applications or novel model development. Compared to [26], this study employs a comprehensive approach by integrating OpenPose with TSR models and SVM to classify walking patterns into skewed or straight forward categories, enhancing clinical diagnostics. This sophisticated system not only develops new statistical models for dynamic and static postural analysis but also focuses on clinical applications, potentially aiding in diagnosing and monitoring mobility impairments like Parkinson’s disease, showcasing a high level of innovation and practical utility in healthcare settings. In summary, while both articles utilize OpenPose, Ref. [26] focuses on a broad validation of OpenPose’s utility in capturing gait parameters, aiming at general advancements in gait analysis technology. In contrast, this study integrates it with other complex models for a targeted clinical application, focusing on diagnosing skewed walking patterns. Viswakumar, Aditya et al. utilize traditional biomechanical tools such as force plates and 3D motion capture to explore how footwear impacts gait dynamics. The findings aim to enhance sports performance and inform rehabilitation strategies by elucidating the biomechanical effects of different footwear. Ref. [27] is less focused on computational models and more on direct biomechanical measurement. In comparison to [27], this study employs sophisticated computational methods to categorize and identify individuals according to their posture and walking patterns. It is closely linked to clinical diagnosis and personalized medicine. The study [28] uses inertial measurement units (IMUs) and machine learning to analyze gait parameters such as speed and symmetry, helping to develop wearable technologies for real-time mobility monitoring, particularly for the elderly. The suggested method emphasizes image-based analysis and computational modeling to assess gait through video data and machine learning, concentrating on intricate posture analysis suitable for clinical diagnostics.

This study utilizing the OpenPose framework and SVM demonstrates a robust approach to gait analysis through video footage, providing non-invasive and advanced methodologies for detecting human keypoints and analyzing gait abnormalities. The proposed approach specifically contributes to the field by developing a TSR model that captures both spatial and temporal dimensions of postural control during walking. This innovative approach allows for a detailed understanding of how posture changes dynamically during movement, providing a more nuanced analysis compared to static measurements typically used in other studies.

### 4.4. Literatures for Health Issues and Postural Control during Walking

Postural control during walking can be indicative of underlying health problems. The body’s postural control significantly impacts the quality of life [29,30]. Using wearable devices to measure posture can be inconvenient for participants [31]. However, this challenge can be addressed by integrating deep learning into Internet of Things (IoT) monitoring systems, enhancing the capability to detect various motions and postures effectively [32]. OpenPose-based deep learning methods have been employed to detect resting tremors and finger tapping [33,34]. Additionally, skeleton normality has been assessed by measuring angles and velocities with these methods [35,36,37]. These techniques are beneficial not only for generating three-dimensional poses [38,39] but also for understanding the correlation between postural behavior and functional disorders such as Parkinson’s disease [40,41], autism spectrum disorder [42], and metatarsophalangeal joint flexions [43]. OpenPose-based deep learning approaches are also instrumental in detecting skeletal, ankle, and foot motion [44], evaluating physical function [45,46], and studying post-stroke conditions [47]. Hence, noninvasive tracking devices are pivotal for recording [48], analyzing, measuring, and detecting body posture, potentially highlighting real-time health issues.

### 4.5. Considerations and Limitations of Video-Based Gait Analysis

The resolution, frame rate, and lighting conditions of video recordings can significantly affect the accuracy of the OpenPose algorithm in detecting joint positions. Videos of lower quality may lead to inaccuracies in joint node plots, potentially impacting the subsequent feature extraction and classification processes. For reliable analysis, video quality requirements include a minimum resolution of 1080p to mitigate issues related to poor video conditions.

Complex gait patterns, such as those involving irregular movements or the use of assistive devices, pose significant challenges for gait analysis. These scenarios can complicate the identification and tracking of joint nodes, leading to potential misclassifications. To enhance the robustness of our model against such variability, we have implemented several strategies, including refining feature extraction techniques and incorporating advanced machine learning strategies to more accurately capture the nuances of complex gait dynamics.

Additionally, we have identified potential research directions aimed at overcoming these limitations. This includes the development of more sophisticated algorithms that are less sensitive to video quality and more adept at handling a broader range of gait abnormalities. Collaborating with biomechanics experts is considered to refine the identification and interpretation of gait patterns, thereby enhancing the overall efficacy and applicability of our analytical methods.

## 5. Conclusions

This study showcased the advantages of the presented method, which included ease of experiment execution, minimal intervention for subjects, task revelation via computer vision schema, and simultaneous qualitative and quantitative analyses. Successful qualitative and quantitative analyses were demonstrated by extracting DJNP using OpenPose deep learning methods, generating essential features with the TSR model, and classifying the three observed walking behaviors—straight forward, skew to left, and skew to right—using the SVM method. The discernable values of CP for TAR and BAR provide rapid differentiation between skew to left and skew to right walking patterns. Moreover, the SVM model developed in this study holds promise as a classifier among groups based on DJNP plots. Future research endeavors will specifically include individuals with gait abnormalities alongside healthy participants. This expansion will allow us to rigorously test and validate the applicability of our proposed method across a broader spectrum of health conditions. By integrating a diverse group with varying gait challenges, we aim to refine our analysis techniques and enhance the method’s utility in real-world clinical settings, ensuring it is robust and effective for diagnosing and managing gait-related disorders.

However, this research has certain limitations. It primarily zeroes in on three specific walking patterns, namely straight forward, skew to left, and skew to right, potentially overlooking other walking posture variations or irregularities. Another limitation is the study’s dependence on video footage quality, frame rate, and resolution. Inaccuracies introduced by low-quality videos or equipment variations could impact posture detection precision. These limitations must be taken into account when interpreting the findings and considering the study’s wider applicability. It is hoped that subsequent research will address these limitations, enhancing both the study’s validity and its relevance.

## Figures and Tables

**Figure 1 bioengineering-11-00548-f001:**
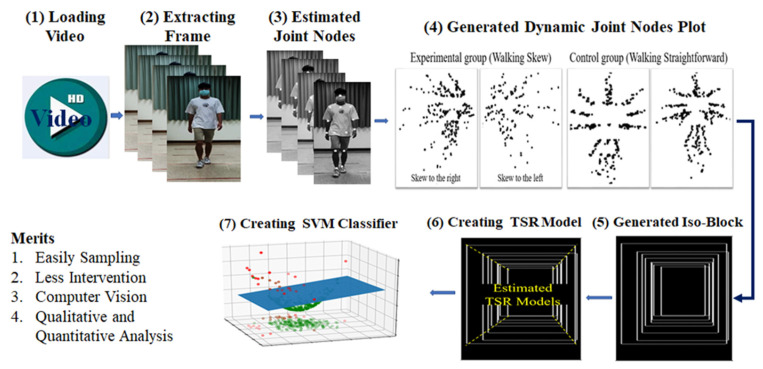
The designed flow of research included loading the video, extracting a single frame, estimating joint nodes by OpenPose deep learning method, generating the iso-block, and creating a temporal and spatial regression model.

**Figure 2 bioengineering-11-00548-f002:**
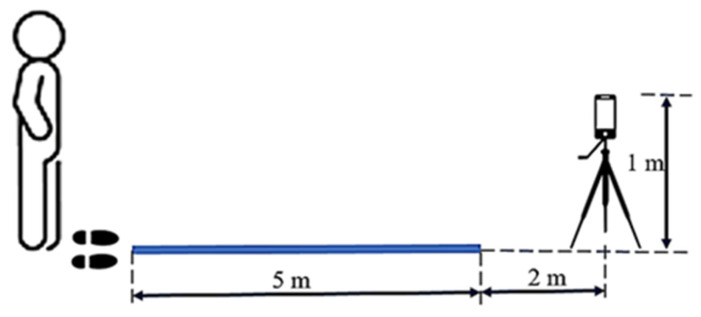
Experimental setup (the cell phone was placed 1 m above the floor and 2 m from the participant).

**Figure 3 bioengineering-11-00548-f003:**
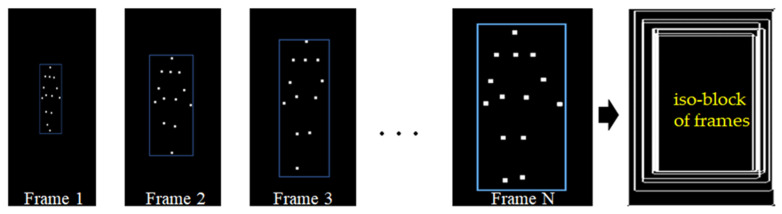
Dynamic joint node plot (DJNP) (right) obtained by merging the heat maps of joint nodes from 1 to N frame by using the OpenPose algorithm.

**Figure 4 bioengineering-11-00548-f004:**
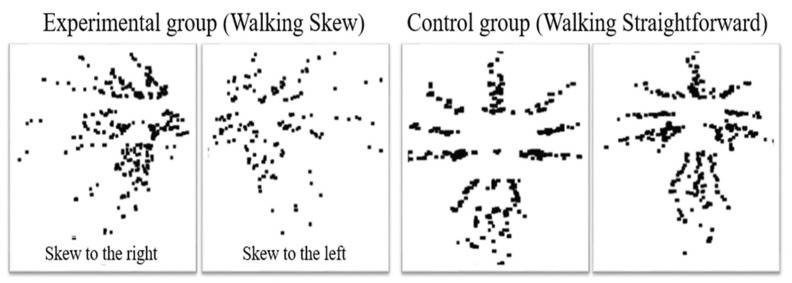
The DJNPs between experimental and control groups were listed.

**Figure 5 bioengineering-11-00548-f005:**
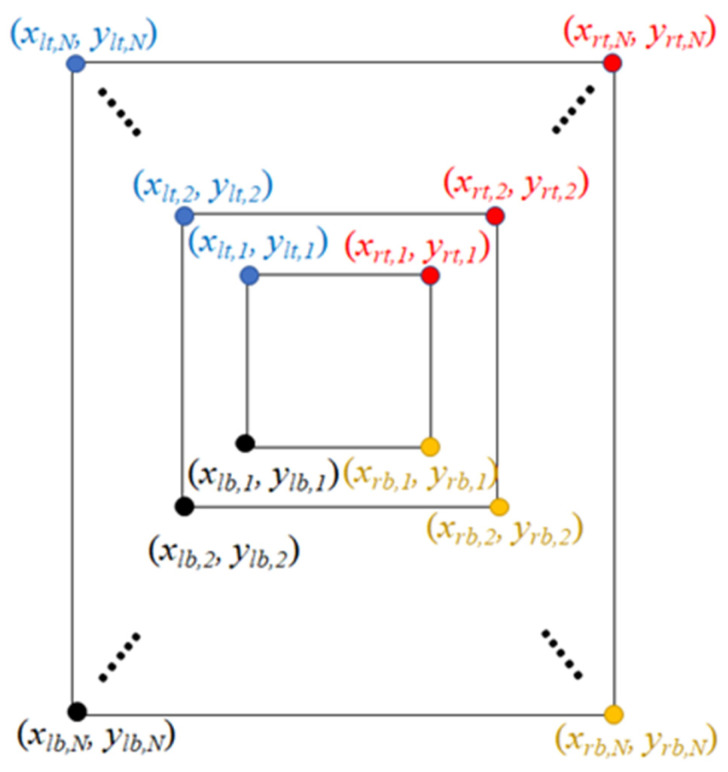
The demography the locations of an iso-block for creating four temporal and spatial regression models.

**Figure 6 bioengineering-11-00548-f006:**
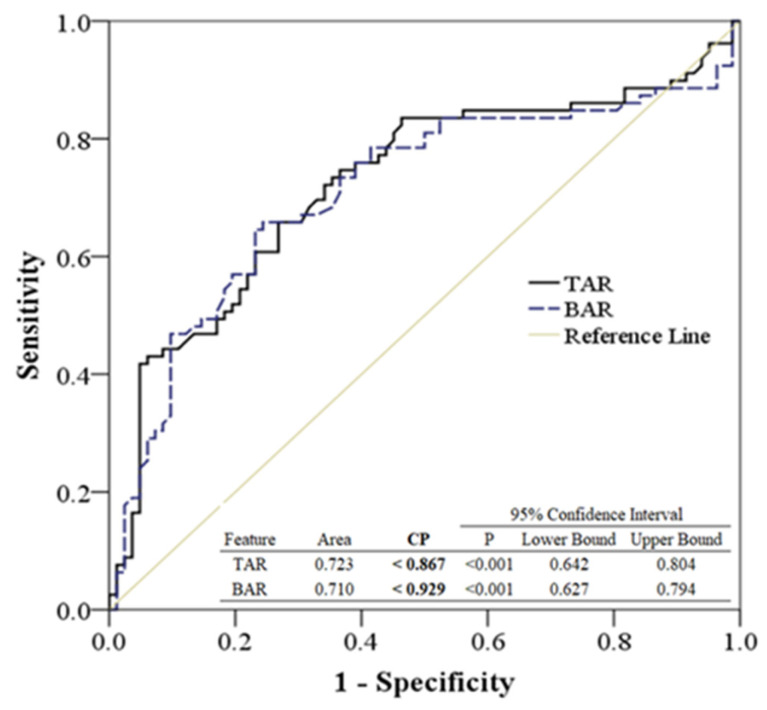
The ROC analysis was generated by TAR and BAR among walking straight forward and skew to right.

**Figure 7 bioengineering-11-00548-f007:**
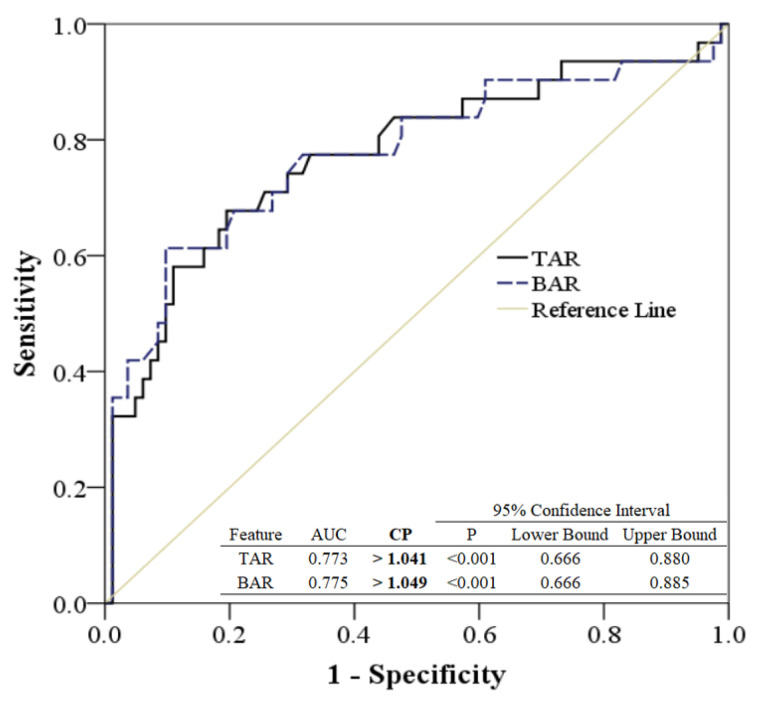
The ROC analysis was generated by TAR and BAR among walking straight forward and skew to left.

**Figure 8 bioengineering-11-00548-f008:**
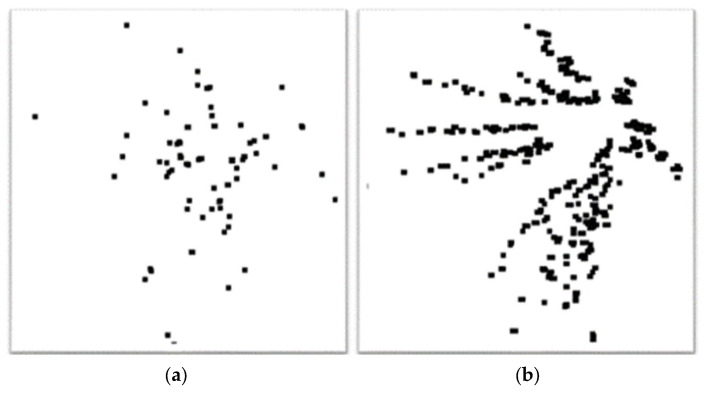
The DJNPs are shown with insufficient and sufficient sampling rates in (**a**) and (**b**), respectively.

**Table 1 bioengineering-11-00548-t001:** The statistics of features among groups with the Kruskal–Wallis test.

Feature	Group	N	Mean	STD	Kruskal–Wallis Test
TAR	Strait	85	0.952	0.147	<0.01
	Skew to Left	36	1.121	0.196	
	Skew to Right	89	0.847	0.175	
BAR	Strait	85	0.965	0.122	<0.01
	Skew to Left	36	1.057	0.105	
	Skew to Right	89	0.918	0.101	
Velocity (m/s)	Strait	85	0.676	0.072	0.882
	Skew to Left	36	0.684	0.085	
	Skew to Right	89	0.694	0.079	

**Table 2 bioengineering-11-00548-t002:** The quantitative analysis among groups with values of CP between TAR and BAR.

Group	TAR	BAR
Skew to Left	<0.867	<0.929
Strait	0.867–1.041	0.929–1.049
Skew to Right	>1.041	>1.049

## Data Availability

Due to privacy and ethical restrictions, the data for this study is unavailable.

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
