# Peer review of "Integrating OpenPose and SVM for Quantitative Postural Analysis in Young Adults: A Temporal-Spatial Approach"

_bioengineering, 2024, doi:10.3390/bioengineering11060548_

Round 1

Reviewer 1 Report

Comments and Suggestions for Authors

Title: Integrating OpenPose and SVM for Quantitative Postural Analysis in Young Adults: A Temporal-Spatial Approach

The manuscript presents an innovative approach of utilizing OpenPose and Support Vector Machine (SVM) for postural analysis in young adults and emphasizes the potential clinical applications of the research. Even though the manuscript is interesting, some aspects should be improved for possible publication and for a better understanding by the readers. As, it lacks specific and technical details regarding the scientific experimental setup and the nature of the walking experiments conducted, which would enhance the reader's understanding of the research study.

Strengths:

  1. The manuscript combines OpenPose for human pose estimation with SVM classification for a novel approach to quantitative postural analysis in walking.
  2. The reported accuracy (0.990) and Kappa index (0.985) suggest excellent performance in classifying walking patterns.
  3. The applicability of the methodology in clinical gait analysis for individuals with mobility limitations is a promising avenue for future research.

Weaknesses:

  1. The authors should give the readers some concrete information to get them excited about their work. The current abstract only describes the general purposes of this research study. It should also include the article's main (1) impact and (2) significance on computational science and machine vision.
  2. The abstract should also state the strength of research results over other at the end followed by some future research directions ae well.
  3. With only 35 participants, the generalizability of the findings is limited. Including a larger and more diverse population could strengthen this research study.
  4. This research study focuses on healthy young adults. Including participants with gait abnormalities would provide a more robust test of the method's effectiveness in real-world scenarios.
  5. While the manuscript mentions using Dynamic Joint Nodes Plots (DJNP), a more detailed explanation of how these plots is used for feature extraction and their contribution to the SVM classification would be beneficial.
  6. A more in-depth explanation of the Temporal and Spatial Regression (TSR) model and the features it extracts would be helpful for understanding the decision-making process of the SVM classifier.
  7. Briefly discussing the exploration of other classification algorithms beyond SVM could strengthen the justification for the chosen method.
  8. While the Area Under the Curve (AUC) values for TAR and BAR are reported, providing context on how they inform the classification performance would be valuable.
  9. Consider including visualizations of DJNP plots and the impact of TAR and BAR on classification.
  10. Discuss limitations of the methodology, such as dependence on video quality and potential challenges in complex gait patterns.
  11. Briefly mention ethical considerations for data collection and participant privacy.
  12. Ensure the clarity and coherence in language and structure throughout the manuscript would enhance readability and facilitate understanding for a wider audience.
Comments on the Quality of English Language

Minor

Reviewer 2 Report

Comments and Suggestions for Authors

It would have been useful to put examples of use and expectations useful for rehabilitating doctors and not simple statements of principle.

Round 2

Reviewer 1 Report

Comments and Suggestions for Authors

Title: Integrating OpenPose and SVM for Quantitative Postural Analysis in Young Adults: A Temporal-Spatial Approach

This manuscript presents a novel and innovative approach for quantitative postural analysis in young adults using OpenPose and Support Vector Machine (SVM). The authors have successfully addressed the comments suggested in the first round of review, resulting in a significantly improved manuscript that is now recommended for possible publication in the current format.

The authors have incorporated the suggestions to strengthen the manuscript in several key areas listed below:

  • The abstract now clearly outlines the research's impact and significance on computational science and machine vision, along with the key findings and future research directions.
  • The authors provide a detailed explanation of Dynamic Joint Nodes Plots (DJNP) and their role in feature extraction for SVM classification.
  • A comprehensive explanation of the Temporal and Spatial Regression (TSR) model and its extracted features enhances understanding of the SVM classifier's decision-making process.
  • The justification for selecting SVM is strengthened by discussing the exploration of alternative classification algorithms.
  • The manuscript incorporates visualizations of DJNP plots and explores the impact of True Acceptance Rate (TAR) and Balanced Accuracy Rate (BAR) on classification performance.
  • Limitations of the methodology, including dependence on video quality and complex gait patterns, are addressed.
  • Ethical considerations for data collection and participant privacy are discussed.

Reviewer 2 Report

Comments and Suggestions for Authors

No comments